# Catalytic asymmetric C–Si bond activation via torsional strain-promoted Rh-catalyzed aryl-Narasaka acylation

Jia Feng[1,3], Xiufen Bi[1,3], Xiaoping Xue[1], Na Li[1], Lei Shi[1] & Zhenhua Gu ⬤ [1,2✉]

Atropisomers are important organic frameworks in bioactive natural products, drugs as well as chiral catalysts. Meanwhile, silanols display unique properties compared to their alcohol analogs, however, the catalytic synthesis of atropisomers bearing silanol groups is challenging. Here, we show a rhodium-catalyzed torsional strain-promoted asymmetric ring-opening reaction for the synthesis of α-silyl biaryl atropisomers. The reaction features a dynamic kinetic resolution of C(Ar)-Si bond cleavage, whose stereochemistry was controlled by a phosphoramidite ligand derived from (S)-3-methyl-1-((2,4,6-triisopropylphenyl)sulfonyl) piperazine. This work is a demonstration of an aryl-Narasaka acylation, where the C(Ar)-Si bond cleavage is promoted by the torsional strain of α, α′-disubstituted silafluorene.

[1] Hefei National Laboratory for Physical Sciences at the Microscale, and Department of Chemistry, Center for Excellence in Molecular Synthesis, University of Science and Technology of China, 96 Jinzhai Road, Hefei, Anhui 230026, PR China. [2] Ocean College, Minjiang University, Fuzhou, Fujian 350108, PR China. [3] These authors contributed equally: Jia Feng, Xiufen Bi. ✉email: zhgu@ustc.edu.cn

Atropisomers are an important class of compounds, which showcased chirality due to the restricted rotation around the single C–C, C–N, or C–O bonds[1–3]. Among them axially chiral biaryls have caught extensive attentions due to their wide existence in bioactive natural products, drugs[4,5]; furthermore it is one of prior skeletons for chiral ligands or catalysts in asymmetric catalysis.

Despite the importance of biaryl atropisomers, the catalytically asymmetric synthesis was challenging due to the steric hindrance around the axis, which usually required three or four ortho substituents for achieving stable axial chirality[6]. The aryl–aryl cross-coupling or oxidative coupling is the most straightforward method, which delivered extremely useful axially chiral ligands and catalysts, such as BINOLs[7–18]. In addition, other useful methods, including (dynamic) kinetic resolution[19–26], point to axial chirality transfer[27–29], de novo aryl ring synthesis[30–34] have been developed and significantly broadened the diversity of biaryl atropisomers. Recently, organo-catalyzed transformation showed its strong power in organic synthesis and it have been successfully applied to axially chiral molecules construction[35–40].

In comparison with the classic asymmetric couplings, which proceed via a highly bulky metal intermediate (Fig. 1a), the ring-opening strategy surcumvented this key intermediate and it provided an efficient protocol for construction of biaryl atropisomers. Bringmann pioneered the atropisomer synthesis via ring-opening of lactones, with either stoichiometric or catalytic methods (Fig. 1b)[41,42]. Recently, Zhang and co-workers realized Ir-catalyzed asymmetric hydrogenation of Bringmann's lactones[43]. Under the catalysis of nickel, copper, and palladium, the groups of Hayashi, Gu and others realized the ring-opening of dibenzo[b,d]thiophenes[44], diaryliodoniums[45–48], and 9H-fluoren-9-ols (Fig. 1b)[49]. This ring-opening reactions showed particular advantages in the preparation of sterically hindered ortho tetra-substituted atropisomeric biaryls; furthermore, these ring-opening reactions displayed excellent diversity: hydroxyl, hydroxymethyl, thiol, iodine, and keto groups were efficiently introduced to the position adjacent to the chiral axis.

Silanols display unique applications in pharmaceutical chemistry and organic chemistry[50,51]. For example, compound 1 displays nuclear receptor modulator activity and has better lipophilicity than the corresponding carbinol[52]. Silanols 2 and 3 were used as organocatalysts for asymmetric syntheses[53,54]. Considering the remarkable property of silanol groups, we wonder if ortho silanol substituted biaryl atropiosmers could be accessed via the carbon–silica bond cleavage of silafluorene in a stereoselective manner. Unfortunately, the silanes showed low reactivity in classic cross-couplings, i.e., Hiyama or Hiyama–Denmark Couplings[55,56]. Pleasingly, the Narasaka acylation[57,58], which favored C–SiMe$_3$ bond breaking over the C–SiMe$_2$(OH), provided a potential solution for C–Si bond cleavage of silafluorene (Fig. 2b). However, challenges still remain: (a) only vinyl silanes underwent Narasaka acylation; (b) it is necessary to differentiate up to four C(aryl)–Si bonds in our systems. The releasing of noncyclic aryl ring is the undesired pathway; (c) the classic Narasaka acylation is steric sensitive, and vinyltriphenylsilane displayed extremely low reactivity (Fig. 2c). It is forseeable, the reaction would be more challenging if tetra(arylsilanes) were used. In the previous studies on the ring-opening reactions, we found that the two non-hydrogen groups adjacent to the axis increased the distortion of the molecule. The distorted molecule is the energetic compound, which had relative lower activation energy than the non-distorted one for the ring-opening reaction. Different from the inherent high strain of three- or four-membered rings, we anticipated that the activity of ring-opening reaction can be increased by the torsional strain of α, α′-disubstituted five-membered silafluorenes. Furthermore, the torsional strain also enabled to differentiate one of four C(aryl)–Si bonds to give desired products.

Here, we report an aryl-Narasaka acylation reaction catalyzed by the chiral rhodium complex which is in situ formed of [Rh(CO)$_2$Cl]$_2$ and phosphoramidite ligand derived from (S)-3-methyl-1-((2,4,6-triisopropylphenyl)sulfonyl)piperazine. Computational studies show that the storioanl stain of 7,7-diphenyl-7H-dinaphtho[2,1-b:1′,2′-d]silole is about 12.58 kcal mol$^{-1}$, which efficiently accelerate the C–Si bond cleavage.

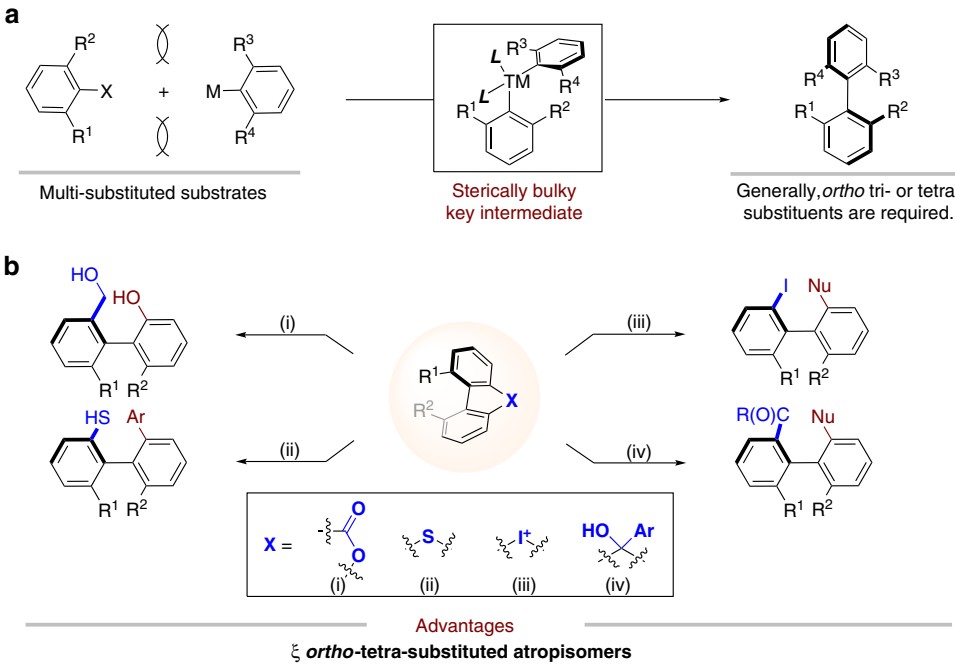

**Fig. 1 Ring-opening strategies for atropisomer synthesis. a** Asymmetric aryl–aryl cross-coupling. **b** Ring-opening strategy for the construction of axially chiral biaryls.

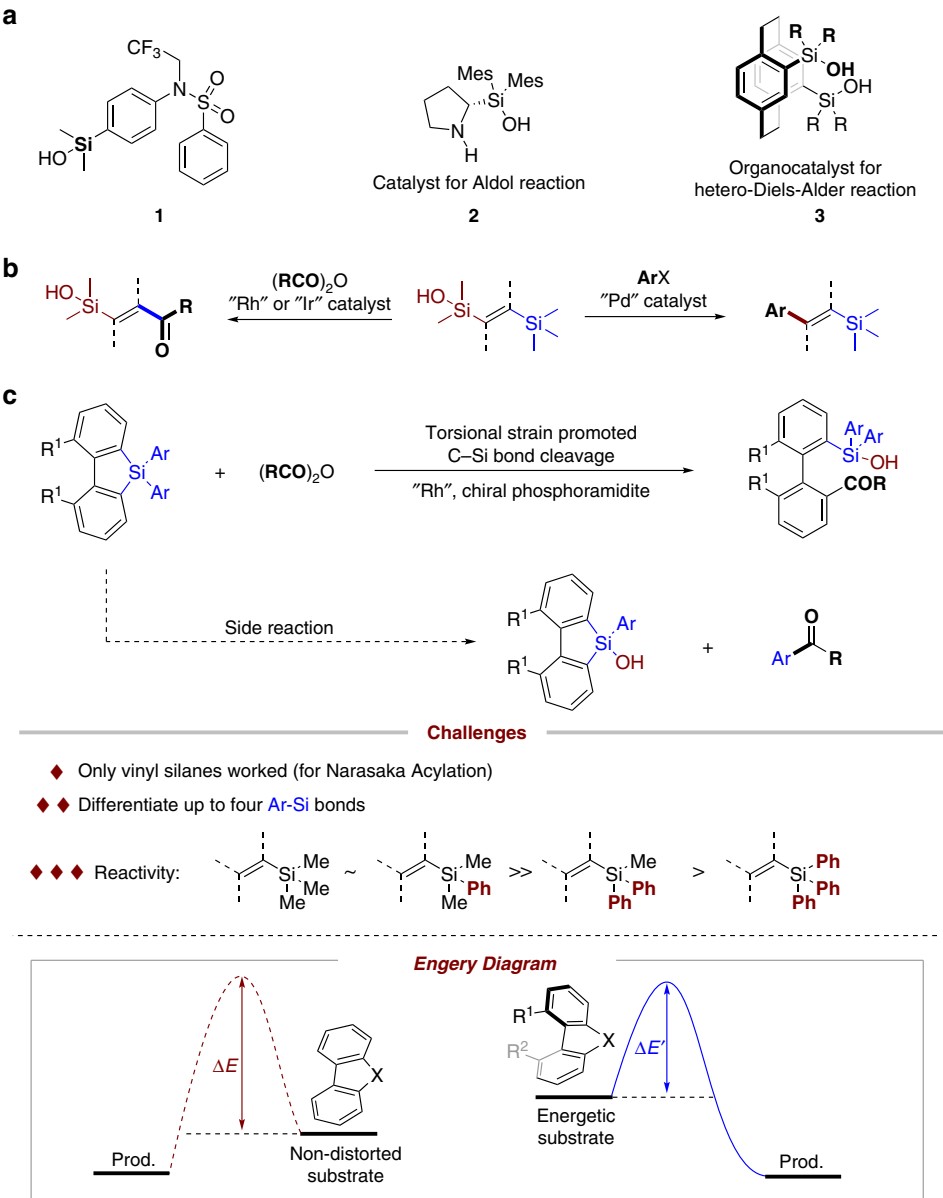

**Fig. 2 Carbon–silica bond cleavage for atropisomer synthesis. a** Silanols in drug scaffold or ligands. **b** Narasaka acylation and Hiyama coupling. **c** Asymmetric Narasaka acylation for atropisomer synthesis.

## Results

**Optimization of Rh-catalyzed aryl-Narasaka acylation.** Our investigation began with the ring-opening/acylation reaction of dimethyl silafluorene (**4a**)[59–62] with acetic anhydride (**5a**) (Table 1). With [Rh(CO)$_2$Cl]$_2$ as the catalyst, primary investigations found that the TADDOL-based phosphoramidite **L1** gave satisfied stereo-induction with 49% yield in the presence of 1.0 equiv of Cs$_2$CO$_3$ (Table 1, entry 1). Decreasing the loading of base gradually improve the yield of **6a**, until it reached the highest level when 30 mol% of Cs$_2$CO$_3$ was employed (Table 1, entries 2–4). Surprisingly, the reaction without the addition of base did not produce isolable desired product (Table 1, entry 5). Replacing the biphenyl group in the ligand with 4-fluorophenyl group gave a decreased enantioselectivity (Table 1, entry 6). The ligand without the chiral α-methyl group at the piperidine moiety (**L3**) afforded a reduced stereoselectivity (Table 1, entry 7). Organic base, i.e., Et$_3$N was not suitable for this ring-opening/acylation reaction, while Na$_2$CO$_3$ dramatically improved the yield and maintained high enantioselectivity (Table 1, entries 8–9). The additives, such as KBr or 4 Å molecular sieve did not offer better outcomes (Table 1, entries 10–11).

The synthesis of sterically hindered silanols is challenging[63,64]. Unfortunately, applying the optimized conditions ([Rh(CO)$_2$Cl]$_2$/ **L1**) to **4a** and benzoic anhydride gave conspicuously decreased ee value (54%, 75% ee). Thus, further optimization based on the modification of phosphoramidites were carried out by using *Si,Si*-diphenylsilafluorene **4b** and benzoic anhydride (Table 2). The *N*-Cbz or *N*-Boc piperazine derived ligands **L4** and **L5** gave improved results in comparison with **L1** (Table 2, entries 1–3). The 2,4,6-trimethylphenylsulfonamide analog **L6** gave 77% ee, while, pleasingly, 2,4,6-tri(isopropyl)phenylsulfonamide derivative **L7** afforded the desired product **6b** in 92% ee (Table 2, entries 4 and 5). The use of Cs$_2$CO$_3$ dramatically improved the yield (Table 2, entry 6). Decreasing the loading of **L7** to 6.25 mol% gave slightly lower yield of **4b** (Table 2, entry 7). Increasing the amount of Cs$_2$CO$_3$ is detrimental for the yield (Table 2, entries 8

**Table 1 Optimization of reaction conditions[a].**

| Entry | Ligand | Base (mol%) | Yield/% | ee/% |
|-------|--------|-------------|---------|------|
| 1 | L1 | Cs$_2$CO$_3$ (100) | 49 | 90 |
| 2 | L1 | Cs$_2$CO$_3$ (80) | 59 | 91 |
| 3 | L1 | Cs$_2$CO$_3$ (50) | 57 | 91 |
| 4 | L1 | Cs$_2$CO$_3$ (30) | 71 | 90 |
| 5 | L1 | - | - | - |
| 6 | L2 | Cs$_2$CO$_3$ (200) | 66 | 86 |
| 7 | L3 | Cs$_2$CO$_3$ (200) | 38 | 81 |
| 8 | L1 | Et$_3$N (30) | - | - |
| 9 | L1 | Na$_2$CO$_3$ (30) | 98 | 90 |
| 10[b] | L1 | Na$_2$CO$_3$ (30) | 88 | 88 |
| 11[c] | L1 | Na$_2$CO$_3$ (30) | 90 | 88 |

[a]Reaction conditions: **4a** (0.10 mmol), **5a** (0.15 mmol, 1.5 equiv), [Rh(CO)$_2$Cl]$_2$ (0.0025 mmol, 2.5 mol%), ligand (0.0105 mmol, 10.5 mol%) and base in 1,4-dioxane at 70 °C for 24 h.
[b]KBr (30 mol%) was added.
[c]30% (w/w) 4 Å MS was added.

and 9). Further optimization by using Et$_3$N in lieu of Cs$_2$CO$_3$, or addition of molecular sieve did not produce better results (Table 2, entries 10 and 11).

**Substrate scope**. Under the optimal conditions, a number of acid anhydrides were tested with **4b** as the substrate (Fig. 3). The aliphatic anhydrides also smoothly coupled with **4b** to produce the corresponding silonols. The longer alkyl chain slightly decreased the yields and selectivity (**6c**–**6f**). 2-Methoxyacetic anhydride steadily underwent this acylation reaction to give **6g** in excellent yield and enantioselectivity. The reaction of cyclopropanecarboxylic anhydride afforded the product **6h** in 81% yield with 85% ee. The acrylic anhydride derivatives also reacted with

**4b** uneventfully to furnish the products in excellent ee values (**6i**–**6k**). The aromatic anhydrides were also compatible substrates. The p-methylbenzoic anhydride gave a slightly decreased yield, while the electron-withdrawing substituents were advantageous for both yields and stereoselectivity (**6l**–**6o**). Subsequently, other substituted aromatic acid anhydrides, including 3,4,5-trimethoxybenzoic anhydride, were submitted to the standard conditions to produce the corresponding silanols **6p**–**6s** with excellent ees. Lastly, the 2-naphthoic or heteroaromatic anhydride were checked, and both of the reactions proceeded smoothly with the ee values were marginally changed (**6t** and **6u**).

Silafluorenes bearing different substituents were further investigated (Fig. 4). Ligand **L1** was found more efficient for substrate **4a**, which reacted with propionic anhydride and butyric

**Table 2 Re-optimization for diphenylsilafluorene[a].**

| Entry | Ligand | Base (mol%) | Yield/% | ee/% |
|-------|--------|-------------|---------|------|
| 1 | **L1** | $Na_2CO_3$ (30) | 41 | 80 |
| 2 | **L4** | $Na_2CO_3$ (30) | 99 | 79 |
| 3 | **L5** | $Na_2CO_3$ (30) | 99 | 86 |
| 4 | **L6** | $Na_2CO_3$ (30) | 99 | 77 |
| 5 | **L7** | $Na_2CO_3$ (30) | 59 | 92 |
| 6 | **L7** | $Cs_2CO_3$ (30) | 94 | 91 |
| 7[b] | **L7** | $Cs_2CO_3$ (30) | 90 | 91 |
| 8 | **L7** | $Cs_2CO_3$ (50) | 79 | 91 |
| 9 | **L7** | $Cs_2CO_3$ (100) | 63 | 91 |
| 10 | **L7** | $Et_3N$ (30) | 25 | 90 |
| 11[c] | **L7** | $Cs_2CO_3$ (30) | 57 | 92 |

[a]Reaction conditions: **4b** (0.10 mmol), **5b** (0.15 mmol, 1.50 equiv), [Rh(CO)$_2$Cl]$_2$ (0.0025 mmol, 2.5 mol%), ligand (0.0105 mmol, 10.5 mol%) and base in 1,4-dioxane at 70 °C for 24 h.
[b]2.5 mol% of [Rh(CO)$_2$Cl]$_2$ and 6.25 mol% of **L7** were used.
[c]30% (w/w) 4 Å MS was added.

anhydride to give **6v** and **6w** in 85% and 90% ee, respectively. The substituted benzene rings attached to the silicon atom have marginal effect on the selectivity, all the silanols were formed in excellent stereoselectivity (**6x**–**6bb**). The silafluorenes with biphenyl skeleton gave relatively lower stereoselectivity (**6cc**–**6ee**). The diastereochemistry of compound **6cc** was determined to be *R* by single crystal X-ray diffraction analysis (CCDC 1968743). The starting material α,α′,β,β′-tetramethyl biphenylsilafluorene

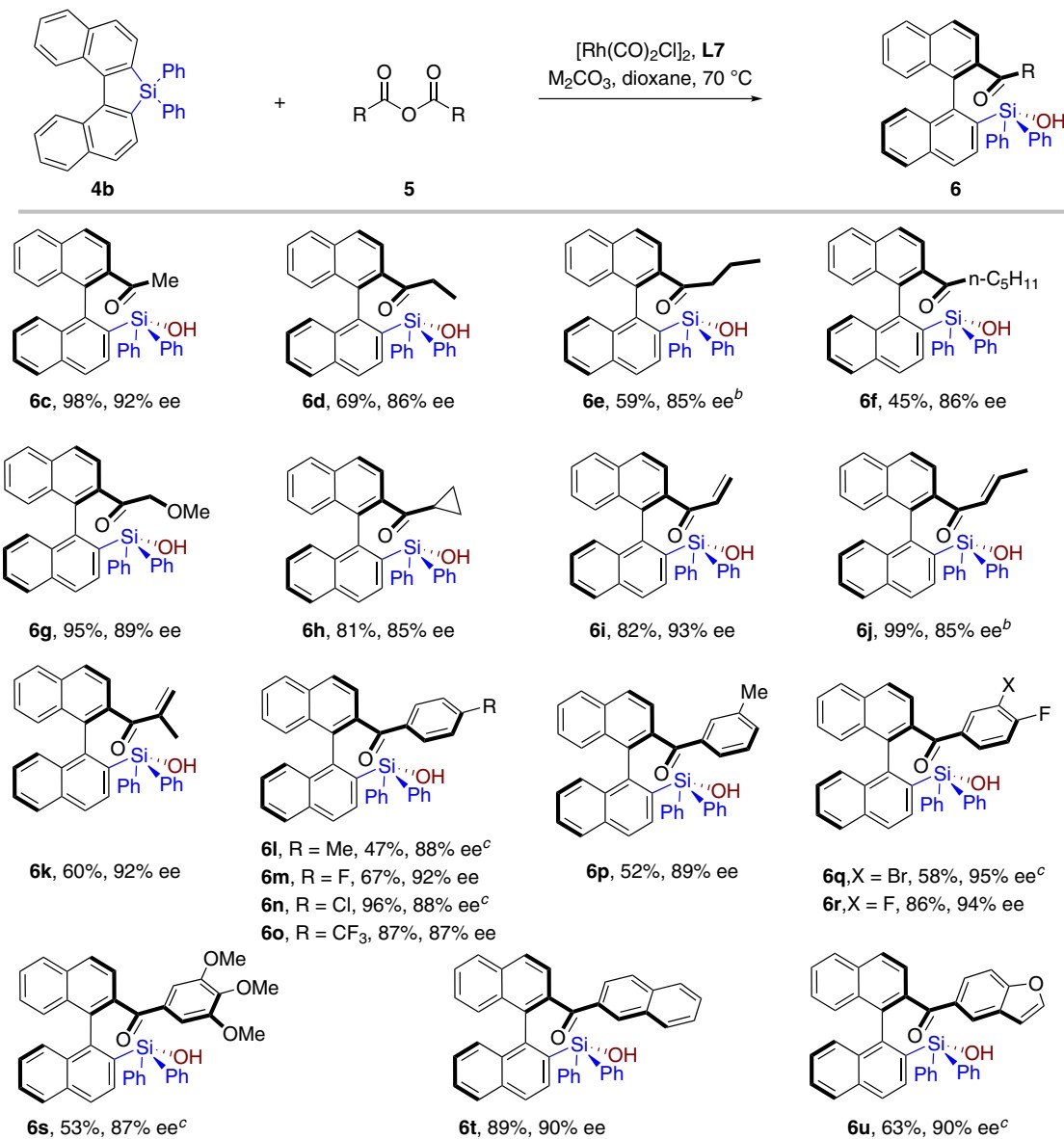

**Fig. 3 Substrate scope[a].** [a]Reaction conditions: silafluorene **4b** (0.20 mmol) and acid anhydrides (0.30 mmol, 1.5 equiv), [Rh(CO)₂Cl]₂ (0.005 mmol, 2.5 mol%), **L7** (0.0125 mmol, 6.25 mol%), and base (30 mol%, Na₂CO₃ for aliphatic anhydride, Cs₂CO₃ for aromatic anhydride) in 1,4-dioxane (4.0 mL) at 70 °C for 24 h. [b]**L1** was used as the ligand. [c]The reaction time is 48 h.

showed slight poor stability, as a result, the corresponding product **6ff** was isolated in only 45% yield with 87% ee.

**Control experiments.** For the unsymmetrical silafluorene **4j**, the reaction gave 1:1 diastereomers, although both **6gg** (CCDC 2013379) and **6gg'** were obtained in high enantioselectivity (Fig. 5a, eq (1)). The reaction between **4b** and benzoic 4-fluorobenzoic anhydride afforded **6m** and **6b** in a ratio of 1:1.3 (Fig. 5a, eq (2)). Notably, with acetic benzoic anhydride as the reagent, the reaction produced methyl aryl ketone **6c** as the major product (**6c:6b** = 6.7:1) (Fig. 5a, eq (3)). Finally, a control reaction by mixing **4a** and **4b** (1:1) was performed with acetic anhydride as limiting reagent. It afforded a 1.5:1 mixture of **6a** and **6c**, indicating **4a** was slightly more active than **4b** under our conditions (Fig. 5b, eq (4)).

**Synthetic application.** Treatment of **6b** with phenylmagnesium chloride at 0 °C afforded hydroxysilanol **7** in excellent yield

(Fig. 5c, eq (5)). Compound **7** was an anologue of corresponding carbols, which was potentially useful in asymmetric catalysis, for example, asymmetric hetero–Diels–Alder reactions[65]. In addition, vinylation of the ketone, followed by bromocyclization gave cyclic compound **9** (CCDC 1987455) in excellent diastereoselectivity (Fig. 5c, eq (6), (7)) (For reaction conditions optimization, see Supplementary Table 1). The Hiyama coupling by employing these silanols was not successful under various conditions, which was possibly due to the steric hindrance of the silanols. However, silanol **6b** was readily transferred to arylbromide **10**, which underwent classic cross-coupling to give **11** without losing the enantiopurity (Fig. 5c, eq (8)).

**Mechanistic studies.** For comparison, compound **4c** was also synthesized, and notably silanol **4c** showed poor reactivity under standard conditions. To get some structural information of these silafluorenes, single crystals of both **4a** (CCDC 1968747) and **4c** (CCDC 1968985) were obtained and analyzed by X-ray diffraction. It clearly shows the biphenyl structure in **4c** is almost planar,

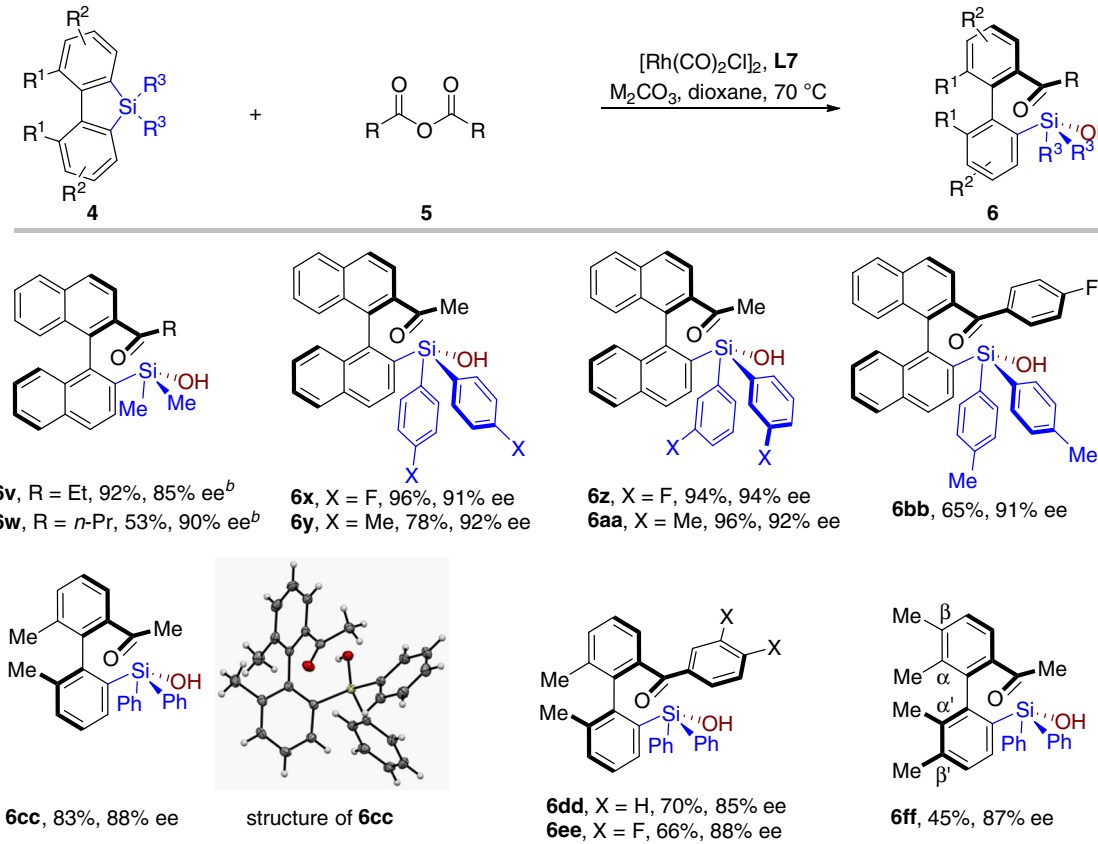

**Fig. 4 Substrate scope[a].** [a]Reaction conditions: silafluorene **4** (0.20 mmol) and acid anhydride **5** (0.30 mmol, 1.5 equiv), [Rh(CO)$_2$Cl]$_2$ (0.005 mmol, 2.5 mol%), **L7** (0.021 mmol, 10.5 mol%) and base (30 mol%, Na$_2$CO$_3$ for aliphatic anhydride, Cs$_2$CO$_3$ for aromatic anhydride) in 1,4-dioxane (4.0 mL) at 70 °C for 24 h. [b]The reaction was performed with **L1** as the ligand at 50 °C for 48 h.

while the increased size of compound **4a** makes the naphthyl rings no longer coplanar. It is a twisted molecule with a distortional angle being 31.8° of the binaphthyl skeleton (Fig. 6a). We calculated the rotational barrier [$\Delta G_{g\,(298.15\,K,\,1\,atm)}$] of **4a**, which is around 20.4 kcal/mol. Thus, the calculated half-life of **4a** is around 94 s at room temperature (Fig. 6b). In order to learn more about the torsional strain energy of **4a**, we calculated the hydrogenation energies of **4a** and **4c**, respectively (Fig. 6c). Considering the structural differences between **4a** and **4c**, we further calculated the hydrogenation energy of noncyclic compounds **4a′** and **4c′** (for details see Supplementary Table 2). Thus, the torsional strain energy of **4a** is calculated as below:

$$\Delta G_{g(298.15\,K,\,1\,atm)} = (\Delta G1 - \Delta G2) - (\Delta G1' - \Delta G2')$$
$$= 12.58\,\text{kcal mol}^{-1}.$$

Markedly, the treatment of [Rh(CO)$_2$Cl]$_2$ with one molar ratio of phosphoramidite **L7** in dichloromethane formed a yellowish dimer [Rh(CO)Cl(**L7**)]$_2$ by releasing two molecules of CO. Recrystallization of this complex in ethyl acetate/hexanes gave a reddish orange crystal that was suitable for X-ray crystallography analysis (CCDC 1969002) (Fig. 6d). In this crystal structure, the (2,4,6-triisopropylphenyl)sulfonylpiperazine moiety worked as a large group by shielding one face of the rhodium center.

Based on the above results and previous studies on Narasaka acylation[66], a brief catalytic cycle was tentatively proposed (Fig. 7). The coordination of **L7** to the pre-catalyst formed monomer Rh(I) complex. The oxidative addition of Rh(I) with silafluorene **4b** cleaved C(Ar)–Si bond to form **10**, which gave optically active biaryl intermediate **11** via reductive elimination forming a Si–O bond. Subsequently, the second oxidation

between **11** with acid anhydride **5** delivered Rh(III) complex **12**. Reductive elimination of **12** furnished acylation product **13**, which would give the final product **6** after hydrolysis of tri(aryl) silyl benzoate moiety.

## Discussion

In conclusion, we reported a Rh-catalyzed asymmetric ring-opening acylation reaction for the synthesis of α-silyl biaryl atropisomers, a class of chiral bulky silanols. The torsional strain of five-membered silafluorenes enabled the success of selective cleavage of C(Ar)–Si bond, thus accomplishing an aryl-Narasaka acylation variant. Additional notable merit of this work is the developed sulfonylpiperazine derived phosphoramidite ligands, which showed high stereo-induction in C(Ar)–Si cleavage/ring-opening reaction.

## Methods

**Typical procedure for the Rh-catalyzed aryl-Narasaka acylation**. Under nitrogen atmosphere, to a Schleck tube was added [Rh(CO)$_2$Cl]$_2$ (2.0 mg, 0.005 mmol, 2.5 mol%), **L1** (24.5 mg, 0.021 mmol, 10.5 mol%) and dioxane (2 mL) at room temperature and was stirred for 30 min. The solution was transferred via cannula carefully to another Schlenk tube charged with **4a** (62 mg, 0.200 mmol, 1.0 equiv), acetic anhydride (28 μL, 0.300 mmol, 1.5 equiv), Na$_2$CO$_3$ (6.4 mg, 0.060 mmol, 30 mol%), and dioxane (2 mL). The tube was capped with a screw cap and stirred at 70 °C for 24 h. After being cooled to room temperature, the mixture was filtered through Celite and the filtrate was concentrated in vacuum and purified by flash column chromatography (PE/EtOAc 90:10) on silica gel to afford **6a** (73 mg, 98%, 90% ee). [α]$_D^{20}$ −2.25 (c 1.13, CH$_2$Cl$_2$). HPLC conditions: Chiralcel AD-H, isopropanol/hexane = 10:90, flow: 1.0 mL/min, λ = 254 nm. **$^1$H NMR** (500 MHz, CDCl$_3$) δ 8.04 (d, J = 9.0 Hz, 1H), 7.99 (d, J = 8.0 Hz, 1H), 7.94 (d, J = 8.5 Hz, 1H), 7.92 (d, J = 8.5 Hz, 1H), 7.87 (d, J = 8.0 Hz, 1H), 7.80 (d, J = 9.0 Hz, 1H), 7.55–7.50 (m, 1H), 7.49–7.44 (m, 1H), 7.31–7.26 (m, 1H), 7.26–7.22 (m, 1H), 7.17 (d, J = 8.5 Hz, 1H), 7.13 (d, J = 8.5 Hz, 1H), 2.70 (s, 1H), 1.94 (s, 3H), 0.13 (s, 3H), −0.47 (s, 3H). **$^{13}$C NMR** (126 MHz, CDCl$_3$) δ 205.2, 141.5, 138.3, 137.9, 137.4, 134.2, 133.7, 133.5, 132.7, 130.4, 128.7, 128.2, 128.0, 127.6, 127.5,

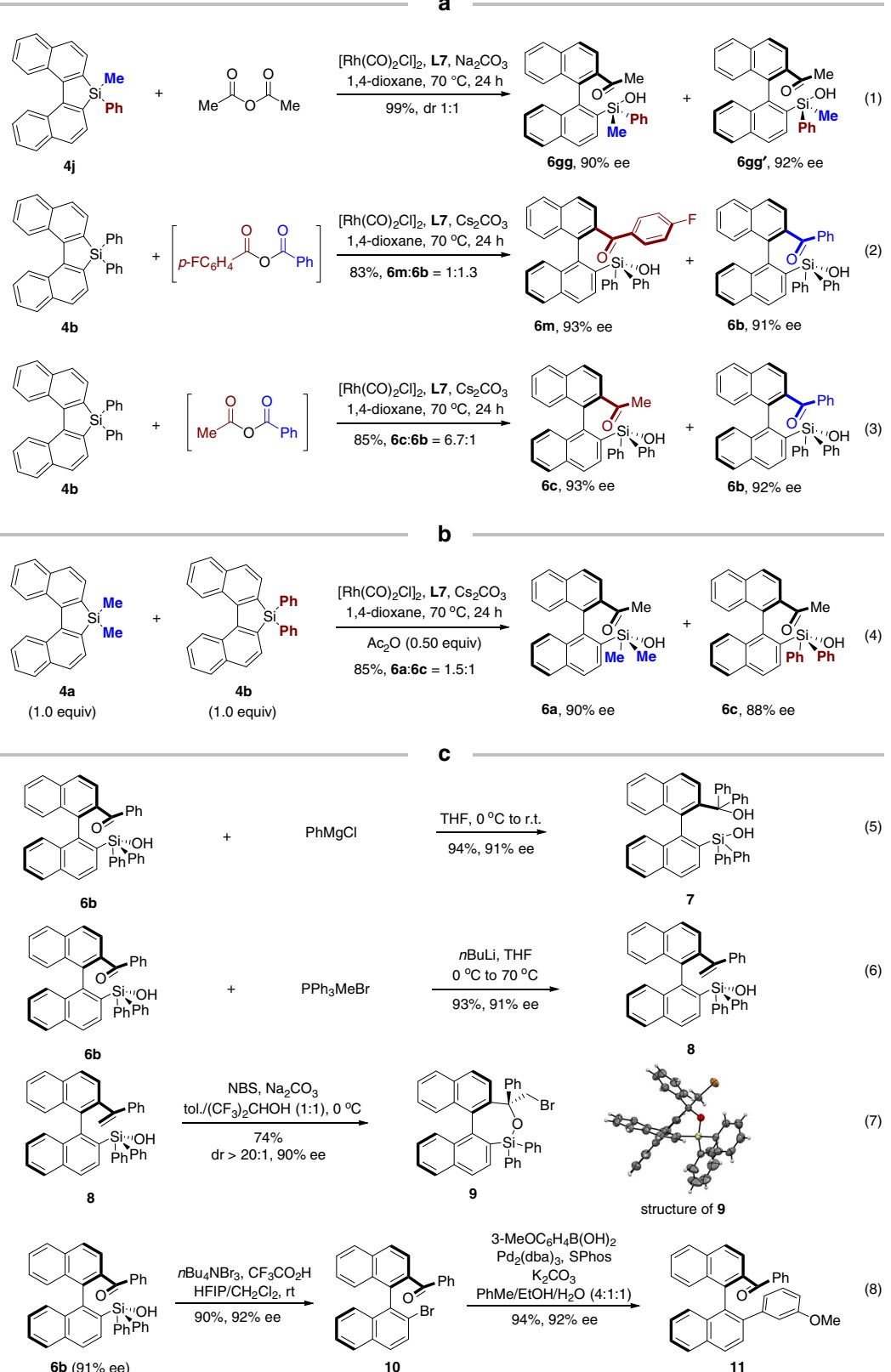

**Fig. 5 Selectivity for unsymmetrical substrates and synthetic applications. a** Reactions of nonsymmetric substrates. **b** Reactivity of different substituted substrates using acetic anhydride as limiting reagent. **c** Synthetic application of axially chiral silanols.

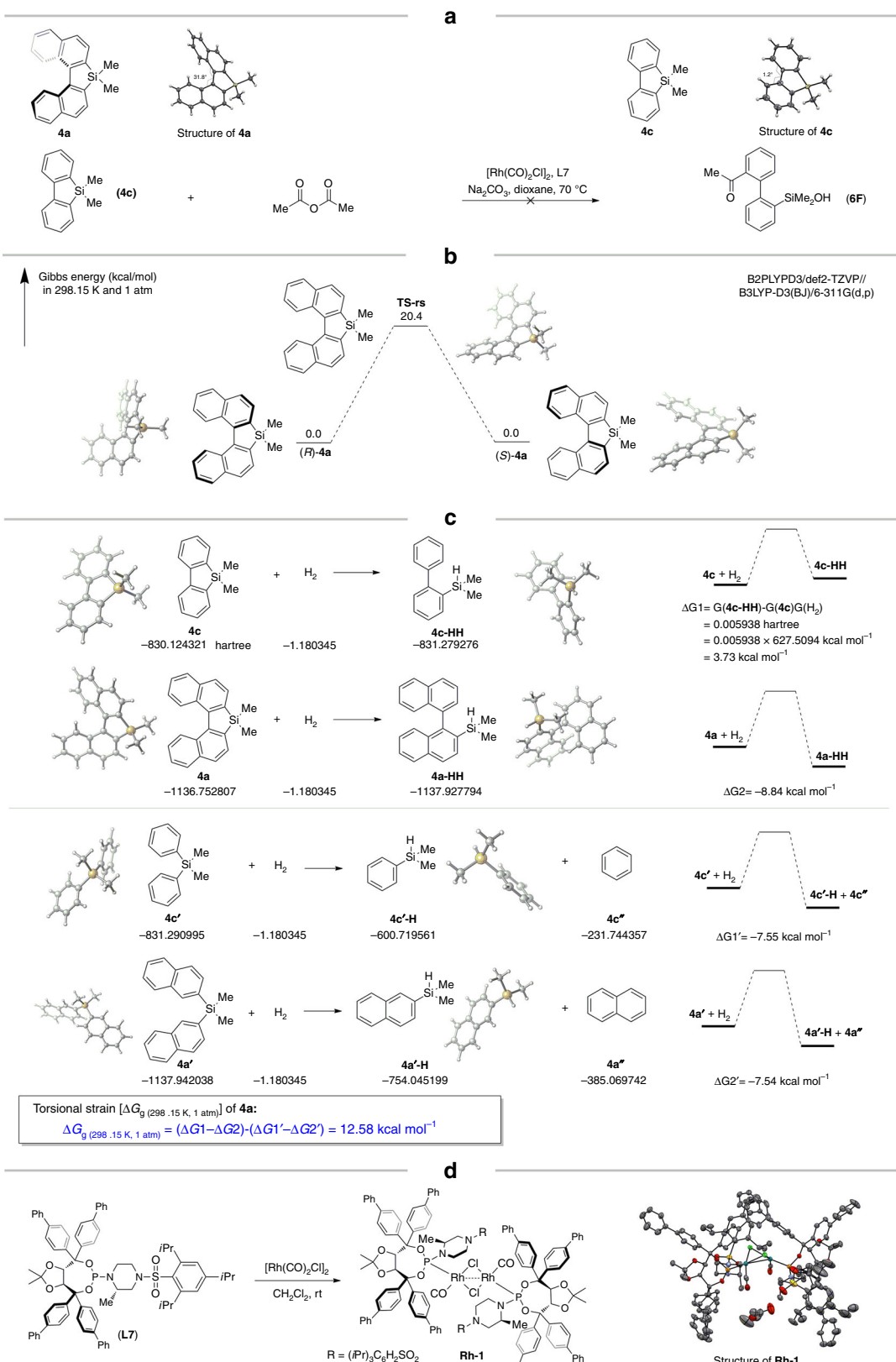

**Fig. 6 Structure and activity relationship. a** Relationship between structure and reactivity of silafluorene. **b** calculated rotational barrier energy of **4a**.
**c** Calculated torsional strain energy of **4a**. **d** Preparation of Rhodium-ligand complex.

**Fig. 7 Possible mechanism.** The oxidative addition of in situ formed Rh(I) catalyst with **4b** gives **10**. Subsequent reductive elimination forms optically active biaryl intermediate **11**. The second oxidation addition of **11** with acid anhydride **5** gives **12**, which regenerates Rh(I) catalyst and compound **6** via reductive elimination, followed by hydrolysis.

127.0, 126.6, 126.5, 126.2, 123.9, 30.5, 0.6. **HRMS (ESI)** calcd for $C_{24}H_{22}O_2SiNa$ [M + Na]$^+$ 393.1287, found 393.1288.

## Data availability

Additional data supporting the findings described in this paper are available in the Supplementary Information. The X-ray crystallographic coordinates for structures reported in this study have been deposited at the Cambridge Crystallographic Data Center (CCDC), under deposition numbers CCDC 1968747 (**4a**), CCDC 16989851 (**4c**), CCDC 1968743 (**6cc**), CCDC 2013379 (**6gg**), CCDC 1969002 (**Rh-1**), and CCDC 1987455 (**9**). These data can be obtained free of charge from the Cambridge Crystallographic Data Center via www.ccdc.cam.ac.uk/data_request/cif.

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

## Acknowledgements

This work was financially supported by National Natural Science Foundation of China (no. 21871241, 21901236), Strategic Priority Research Program of the Chinese Academy of Sciences (XDB20000000), and the Fundamental Research Funds for the Central Universities (WK2060190086). We thank the Supercomputing Center of University of Science and Technology of China for providing the computational resources.

## Author contributions

J.F. designed the approach and performed the experiments, analyzed the experimental data, and prepared the Supplementary Information. X.B. modified the reaction conditions, expanded the scope of the substrates. J.F. and X.B. contributed equally to this project. X.X. performed the calculation. N.L. and L.S. were involved in the preparation of substrates. Z.G. directed the investigations and prepared the paper.

## Competing interests

The authors declare no competing interests.
