## [Peer Review File · Nature Communications]

REVIEWER COMMENTS

Reviewer #1 (Remarks to the Author):

The authors developed a rhodium-catalyzed asymmetric ring-opening reaction of cyclized silfluorenes for the synthesis of α -silyl biaryl atropisomers. The reaction featured a dynamic kinetic resolution of C(Ar)-Si bond cleavage, and the stereochemistry was controlled by chiral phosphoramidite ligand. This method is novel and has potential applications in preparation of silanol-containing chiral ligands. However, major revision is required before it is suitable for publication in Nature Communication.

- 1) Line 11-12 and fig 2a: The products obtained by this method are axially biaryls bearing with specific silanol functional groups. Hence, the authors should focus on the importance of silanols other than the general Si-containing compounds. The sentences in line 11-12 need to be rephrased, and the structures in fig 2a should contain only silanol containing compounds.
- 2) Line 67-70 and fig 2c: The authors compared the reactivities of different vinyl silanes and drew the energy diagram, which need to be supported by references or DFT calculation results. In line 60, the author stated that the reaction would be more challenging if tetra(arylsilanes) were used than the less steric one. However, it turns out the tetra(arylsilanes) generally gave better results than the dimethylsilafluorene. The authors need to explain the difference between these two types of substrates.
- 3) The author conducted the condition optimization with dimethylsilafluorene 4b as the model substrate. However, there is only one dimethylsilafluorene example in the substrate scope tables. I speculate that those dimethylsilafluorene substrates tested by the authors didn't give good results. That's why they switched to the tetra(arylsilanes). The authors need to give an explanation. Also more than half of the ees are below 90%, ideally the condition optimization table need to be done again with 4b as the model substrate, which would make the condition optimization table reasonable, also it might lead to better ee results.
- 4) The reaction conditions in the whole manuscript are confusing. For example, the ratio between Rh and ligand are variable (2.5:10.5 in table 1; 1:1.25 in fig 3; 2.5:6.25 in fig 4; 2.5:10.5 in fig.5). Basically, there is no standard optimized conditions suitable for most of the substrates.
- 5) With aromatic anhydride as the substrate, Na₂CO₃ was used as base in the optimized condition (Fig 3). However, Cs₂CO₃ was used as base in the substrate scope tables. The results and descriptions are not consistent.
- 6) The products 6A-6E in fig 5 need to be re-labelled.
- 7) The authors need to test at least one example for unsymmetrical silafluorene substrate and mixed anhydride respectively to demonstrate the regioselectivity of this reaction.
- 8) For the synthetic application, the authors need to demonstrate at least one example of Hiyama coupling.
- 9) The whole manuscript needs to be further modified. Following are some mistakes:
Line 20: unpreudent should be unprecedent;
Fig. 2c, "C-Si bond cleavage" should be "C-Si bond cleavage".
Fig. 8, ligand "L7" should be bold.
Ref 44, "1, 1'-Binaphthyls" should be "1,1'-Binaphthyls";
Ref 54, "drug saffolds" should be "drug scaffolds".
In SI, typical procedure B, please correct the structure of dimethyl cyclic diaryliodonium.
In SI, compound 7 (91%ee should be 91% ee), 8 (91%ee should be 91% ee), 9 (90%ee should be 90% ee)
In SI, some NMR are missing (compound 4c, 6t); 19F NMR for 6r is strange.
- 8) The complete original HPLC spectrum should be provided.

Reviewer #2 (Remarks to the Author):

The present manuscript submitted by Gu describes a rhodium-catalyzed reaction of silafluorenes with acid anhydrides. Axially-chiral biaryl ketones having a silanol functionality are enantioselectively synthesized. Torsional strain of the silafluorene skeleton is a key requisite for the reaction to occur. Whereas the two phenyl rings of a biphenyl-type silafluorene can take a planar arrangement, torsional strain arises between the two naphthyl rings of a binaphthyl-type silafluorene, which has to take a twisted conformation. The former lacks the reactivity whereas the latter successfully react. The manuscript presents an interesting example of a reaction which is driven by the release of the torsional strain inherent in fused biaryls. I feel such reactions have attracted little attention from a synthetic viewpoint. In addition, a good enantioselectivity is observed. The produced axially chiral biaryl-substituted silicon compounds are potentially useful but otherwise difficult to synthesize. I recommend that the present manuscript be accepted for publication.

1. I wonder if it is possible to estimate the strain energy of the binaphthyl-type silafluorene when compared with that of the biphenyl-type silafluorene.
2. Silanols serve as valuable precursors of the Hiyama-type cross-coupling reaction. Is it possible to use the present products for the Hiyama-type cross-coupling reaction?
3. I wonder how fast the binaphthyl-type silafluorene racemize. Or it is rigid and one of the two Si-C bonds is selectively broken?

We would like to thank all the reviewers for your time and efforts in evaluating our manuscript (Manuscript ID: NCOMMS-20-12289). We have made all of the requested changes, which are listed point-by-point in this letter.

Reviewer 1:

1. **Comment:** The authors developed a rhodium-catalyzed asymmetric ring-opening reaction of cyclized silfluorenes for the synthesis of α -silyl biaryl atropisomers. The reaction featured a dynamic kinetic resolution of C(Ar)-Si bond cleavage, and the stereochemistry was controlled by chiral phosphoramidite ligand. This method is novel and has potential applications in preparation of silanol-containing chiral ligands. However, major revision is required before it is suitable for publication in Nature Communication.

Answer: We thank the reviewer's comments.

2. **Comment:** Line 11-12 and fig 2a: The products obtained by this method are axially biaryls bearing with specific silanol functional groups. Hence, the authors should focus on the importance of silanols other than the general Si-containing compounds. The sentences in line 11-12 need to be rephrased, and the structures in fig 2a should contain only silanol containing compounds.

Answer: (a) The discussion on the utilities of silicon was deleted. Thus, the corresponding references were deleted.

(b) In Fig 2a, only silanols were presented. Thus, two new silanols **2** and **3** were added to Fig. 2a, and the corresponding references were added.

a) Silanols in Drug Scaffold or Catalysts

3. **Comment:** Line 67-70 and fig 2c: The authors compared the reactivities of different vinyl silanes and drew the energy diagram, which need to be supported by references or DFT calculation results. In line 60, the author stated that the reaction would be more challenging if

tetra(arylsilanes) were used than the less steric one. However, it turns out the tetra(arylsilanes) generally gave better results than the dimethylsilafluorene. The authors need to explain the difference between these two types of substrates.

Answer: (a) We performed the calculation on the torsional strain of compound **4a**. The details are listed below. The calculated torsion strain ($\Delta G_{(298.15\text{K})}$) is 12.58 kcal mol⁻¹.

We calculate the torsional strain energy by hydrogenating **4a** and **4c**, respectively. Considering the structural differences between **4a** and **4c**, we further calculated the hydrogenation energy of non-cyclic compounds **4a'** and **4c'**. Thus, the torsional strain energy of **4a** is:

$$\Delta G_{\text{g}}(298.15 \text{ K}, 1 \text{ atm}) = (\Delta G1 - \Delta G2) - (\Delta G1' - \Delta G2') = 12.58 \text{ kcal mol}^{-1}$$

Torsional Strain [$\Delta G_{\text{g}}(298.15 \text{ K}, 1 \text{ atm})$] of **4a**:

$$\Delta G_{\text{g}}(298.15 \text{ K}, 1 \text{ atm}) = (\Delta G1 - \Delta G2) - (\Delta G1' - \Delta G2') = 12.58 \text{ kcal mol}^{-1}$$

(b) The reactivity between distorted cyclic compounds and non-distorted compounds have been calculated in our previous reports on the Cu-catalyzed amination of cyclic diaryliodoniums (*Chem*, **2018**, *4*, 599).

(c) The sequence of the reactivity between different silanes was from the literature.

Entry	Si	time/h	yield/%	recovery of vinylsilane
1	SiMe ₃	18	82	-
2	SiMe ₂ Ph	7	84	-
3	SiMePh ₂	30	71	8
4	SiPh ₃	40	21	64

Yamane, M.; Uera, K.; Narasaka, K. *Bull. Chem. Soc. Jpn.* **78**, 477-486 (2005)

The yields of two independent reactions do not reflect the reactivities of two different substrates. For comparison, a control reaction by mixing **4a** and **4b** (1:1) was performed with acetic anhydride as limiting reagent. The reaction afforded **6a** in 51% yield and **6c** in 34% yield. It indicated that **4a** was slightly more active than **4b** under our conditions.

These results have been added to the text.

4. **Comment:** The author conducted the condition optimization with dimethylsilafluorene **4b** as the model substrate. However, there is only one dimethylsilafluorene example in the substrate scope tables. I speculate that those dimethylsilafluorene substrates tested by the authors didn't gave good results. That's why they switched to the tetra(arylsilanes). The authors need to give an explanation. Also more than half of the ees are below 90%, ideally the condition optimization table need to be done again with **4b** as the model substrate, which would make the condition optimization table reasonable, also it might lead to better ee results.

Answer: (a) Yes, the reviewer's comments are right. Only aliphatic acid anhydrides were suitable coupling components for dimethylsilafluorene **4a**. For diphenylsilafluorene **4b**,

aromatic acid anhydrides are good coupling components.

(b) We added another substrate of aliphatic acid anhydride $[(n\text{-C}_3\text{H}_7\text{CO})_2\text{O}]$. The reaction gave 53% yield and 90% ee.

(c) The optimization table for diphenylsilafluorene **4b** was added to the text.

Entry	Ligand	Base (mol%)	Yield/%	ee/%
1	L1	Na ₂ CO ₃ (30)	41	80
2	L4	Na ₂ CO ₃ (30)	99	79
3	L5	Na ₂ CO ₃ (30)	99	86
4	L6	Na ₂ CO ₃ (30)	99	77
5	L7	Na ₂ CO ₃ (30)	59	92
6	L7	Cs ₂ CO ₃ (30)	94	91
7 ^b	L7	Cs ₂ CO ₃ (30)	90	91
8	L7	Cs ₂ CO ₃ (50)	79	91

9	L7	Cs ₂ CO ₃ (100)	63	91
10	L7	Et ₃ N (30)	25	90
11 ^c	L7	Cs ₂ CO ₃ (30)	57	92

^a Reaction Conditions: **4b** (0.10 mmol), **5b** (0.15 mmol, 1.50 equiv), [Rh(CO)₂Cl]₂ (0.0025 mmol, 2.5 mol%), ligand (0.0105 mmol, 10.5 mol%) and base in 1,4-dioxane at 70 °C for 24 h. ^b 2.5 mol% of [Rh(CO)₂Cl]₂ and 6.25 mol% of **L7** were used. ^c 30% (w/w) 4Å MS was added.

5. **Comment:** The reaction conditions in the whole manuscript are confusing. For example, the ratio between Rh and ligand are variable (2.5:10.5 in table 1; 1:1.25 in fig 3; 2.5:6.25 in fig 4; 2.5:10.5 in fig.5). Basically, there is no standard optimized conditions suitable for most of the substrates.

Answer: We thank this pointing out this confused reaction conditions.

- (a) Generally, the reaction was performed under 2.5 mol% of [Rh(CO)₂Cl]₂ and 10.5 mol% of ligand. However, in most of the cases, good yields and enantioselectivity had marginal effect when 2.5 mol% of [Rh(CO)₂Cl]₂ and 6.25 mol% of ligand were used.
- (b) In Fig. 3, Rh/L = 1:1.25 means 2.5 mol% of [Rh(CO)₂Cl]₂ and 6.25 mol% of ligand.

We have corrected this inaccurate statement. “Rh/L = 1:1.25” has been changed to “2.5 mol% of [Rh(CO)₂Cl]₂ and 6.25 mol% of **L7**”.

6. **Comment:** With aromatic anhydride as the substrate, Na₂CO₃ was used as base in the optimized condition (Fig 3). However, Cs₂CO₃ was used as base in the substrate scope tables. The results and descriptions are not consistent.

Answer: A proper reaction conditions optimization table (Table 2, revised version) has been added to the text.

Entry	Ligand	Base (mol%)	Yield/%	ee/%
1	L1	Na ₂ CO ₃ (30)	41	80
2	L4	Na ₂ CO ₃ (30)	99	79
3	L5	Na ₂ CO ₃ (30)	99	86
4	L6	Na ₂ CO ₃ (30)	99	77
5	L7	Na ₂ CO ₃ (30)	59	92
6	L7	Cs ₂ CO ₃ (30)	94	91
7 ^b	L7	Cs ₂ CO ₃ (30)	90	91
8	L7	Cs ₂ CO ₃ (50)	79	91
9	L7	Cs ₂ CO ₃ (100)	63	91
10	L7	Et ₃ N (30)	25	90
11 ^c	L7	Cs ₂ CO ₃ (30)	57	92

^a Reaction Conditions: **4b** (0.10 mmol), **5b** (0.15 mmol, 1.50 equiv), [Rh(CO)₂Cl]₂ (0.0025 mmol, 2.5 mol%), ligand (0.0105 mmol, 10.5 mol%) and base in 1,4-dioxane at 70 °C for 24 h. ^b 2.5 mol% of [Rh(CO)₂Cl]₂ and 6.25 mol% of **L7** were used. ^c 30% (w/w) 4Å MS was added.

7. **Comment:** The products 6A-6E in fig 5 need to be re-labelled.

Answer: The products 6A-6E in Fig 5 were re-labelled as “6aa-6ee”.

8. **Comment:** The authors need to test at least one example for unsymmetrical silafluorene substrate and mixed anhydride respectively to demonstrate the regioselectivity of this reaction.

Answer: We have tested an unsymmetrical silafluorene substrate. The reaction gave a mixture of diastereomers with dr ratio being 1:1, while the enantioselectivity of the two diastereomers was still good.

We have tested two mixed anhydrides. One is aromatic/aromatic mixed anhydride, and the other is aliphatic/aromatic mixed anhydride. With benzoic 4-fluorobenzoic anhydride, the reaction afforded **6m:6b** in a ratio of 1:1.3. With acetic benzoic anhydride, acetyl compound **6c** was formed as the major product.

9. **Comment:** For the synthetic application, the authors need to demonstrate at least one example of Hiyama coupling.

Answer: (a) For our substrates, the Hiyama coupling did not work in various conditions. This is possibly due to the steric hindrance for the biaryl silanols. In Denmark's studies,

with TBAF as base, there was only a modest steric effect (except for tert-butyl substituents); in the presence of TMSOK, a significant steric effect was noted (Denmark, S. E.; Neuville, L.; Christy, M. E. L.; Tymonko, S. A. *J. Org. Chem.* **2006**, *71*, 8500-8509).

In the trials of Hiyama couplings, various conditions were tried, including pre-deprotonation to prepare silanolate with strong bases. However, either desilylation/protonation product or disilyl ether was detected in these reactions. No desired coupling product could be formed.

Entry	ArX	[Pd] (10%)	ligand	base (2.0 eq)	additive (2.0 eq)	solvent	temp./°C	Outcome
1	4-MeOC ₆ H ₄ Br	Pd ₂ (dba) ₃	--	TBAF	CuI	THF	60	b1
2	4-MeC ₆ H ₄ I	Pd(PPh ₃) ₄	--	--	Ag ₂ O	THF	60	b1
3	4-MeOC ₆ H ₄ Br	[Pd(allyl)Cl] ₂	dppb	Cs ₂ CO ₃	H ₂ O	Toluene	100	NR
4	4-MeOC ₆ H ₄ Br	Pd ₂ (dba) ₃	Sphos	NaH	--	THF	70	b1 & b2
5	4-MeOC ₆ H ₄ Br	Pd ₂ (dba) ₃	--	KOTMS	--	Dioxane	rt	NR
6	4-CF ₃ C ₆ H ₄ Br	Pd ₂ (dba) ₃	Sphos	KOtBu & KHMDS	--	Toluene	90	Complex
7	4-MeOC ₆ H ₄ Br	Pd ₂ (dba) ₃	Sphos	KOtBu	18-Crown-6	THF	70	b1
8	4-CF ₃ C ₆ H ₄ I	Pd ₂ (dba) ₃	--	NaOtBu	Cu(OAc) ₂	Toluene	rt	b1
9	4-MeCOC ₆ H ₄ I	[Pd(allyl)Cl] ₂	P(O)Ph ₃	--	--	Toluene	90	NR
10	4-MeCOC ₆ H ₄ I	[Pd(allyl)Cl] ₂	--	--	KF	DMF	120	b1
11	2-BrPyridine	PdCl ₂ (PPh ₃) ₂	PPh ₃	TBAF	CuI	DMF	rt	b1

Entry	Base	ArX	Condition	Outcome
1	KHMDS	4-EtO ₂ CC ₆ H ₄ I	[Pd(allyl)Cl] ₂ , tBu ₃ PHBF ₄ , Toluene, 90 °C	b1
2	NaHMDS	4-EtO ₂ CC ₆ H ₄ I	[Pd(allyl)Cl] ₂ , tBu ₃ PHBF ₄ , Toluene, 90 °C	b1
3	KHMDS	4-EtO ₂ CC ₆ H ₄ I	Pd ₂ (dba) ₃ , Sphos, THF, 70 °C	b1
4	KH	4-MeOC ₆ H ₄ Br	Pd ₂ (dba) ₃ , Sphos, THF, 70 °C	b1

(b) Besides the utility of silanols in pharmaceutical chemistry and asymmetric catalysis, the silanol group was readily transferred to the bromide, which could be used in classic cross-couplings.

10. **Comment:** The whole manuscript needs to be further modified. Following are some mistakes:

Line 20: unpreudent should be unprecedented;

Fig. 2c, “C-Si bond cleavage” should be “C-Si bond cleavage”.

Fig. 8, ligand “L7” should be bold.

Ref 44, “1, 1’-Binaphthyls” should be “1,1’-Binaphthyls”;

Ref 54, “drug saffolds” should be “drug scaffolds”.

In SI, typical procedure B, please correct the structure of dimethyl cyclic diaryliodonium.

In SI, compound **7** (91%ee should be 91% ee), **8** (91%ee should be 91% ee), **9** (90%ee should be 90% ee)

In SI, some NMR are missing (compound 4c, 6t); ^{19}F NMR for 6r is strange.

Answer: (a) The typos have been corrected.

(b) Structure of cyclic diaryliodonium has been corrected.

(c) “91%ee” has been changed to “91% ee”; “91%ee” has been changed to “91% ee”; “90%ee” has been changed to “90% ee”;

(d) The missing NMR spectra have been added.

(e) ^{19}F NMR of **6r** has been re-analyzed.

11. **Comment:** The complete original HPLC spectrum should be provided.

Answer: The complete original HPLC traces were added to the Supporting Information.

Reviewer 2:

1. **Comment:** The present manuscript submitted by Gu describes a rhodium-catalyzed reaction of silafluorenes with acid anhydrides. Axially-chiral biaryl ketones having a silanol functionality are enantioselectively synthesized. Torsional strain of the silafluorene skeleton is a key requisite for the

reaction to occur. Whereas the two phenyl rings of a biphenyl-type silafluorene can take a planar arrangement, torsional strain arises between the two naphthyl rings of a binaphthyl-type silafluorene, which has to take a twisted conformation. The former lacks the reactivity whereas the latter successfully react. The manuscript presents an interesting example of a reaction which is driven by the release of the torsional strain inherent in fused biaryls. I feel such reactions have attracted little attention from a synthetic viewpoint. In addition, a good enantioselectivity is observed. The produced axially chiral biaryl-substituted silicon compounds are potentially useful but otherwise difficult to synthesize. I recommend that the present manuscript be accepted for publication.

Answer: We thank the reviewer's comments.

2. **Comment:** I wonder if it is possible to estimate the strain energy of the binaphthyl-type silafluorene when compared with that of the biphenyl-type silafluorene.

Answer: We thank the reviewer pointing out this important aspect regarding our work. The Reviewer 1 also raised a similar question. Thus, I discuss the calculation results here again.

We calculate the torsional strain energy by hydrogenating **4a** and **4c**, respectively. Considering the structural differences between **4a** and **4c**, we further calculated the hydrogenation energy of non-cyclic compounds **4a'** and **4c'**. Thus, the torsional strain energy of **4a** is:

$$\Delta G_{g(298.15\text{ K}, 1\text{ atm})} = (\Delta G1 - \Delta G2) - (\Delta G1' - \Delta G2') = 12.58\text{ kcal mol}^{-1}$$

3. **Comment:** Silanols serve as valuable precursors of the Hiyama-type cross-coupling reaction. Is it possible to use the present products for the Hiyama-type cross-coupling reaction?

Answer: We thank this reviewer's comments. The Hiyama coupling was unsuccessful in various conditions, which was possibly attributed to the steric hindrance of the silanols. However, the silyl group was efficiently transferred to arylbromide, which could undergo classic cross-couplings, such as Suzuki coupling.

4. **Comments:** I wonder how fast the binaphthyl-type silafluorene racemize. Or it is rigid and one of

the two Si–C bonds is selectively broken?

Answer: (a) We calculated the rotational barrier of the two conformers. The rotational barrier (ΔG) of **4a** is 20.4 kcal mol⁻¹. Thus, configurationally dynamic with a half-life is around 94 seconds under 1 atmosphere at 298.15 K. It means that the silafluorenes undergo fast equilibrium at the standard reaction conditions at 70 °C (343.15 K).

We thank all reviewers for their valuable comments that we believe substantially improved the quality of our manuscript.

With best regards,

Zhenhua Gu, Ph.D.
Professor of Chemistry
Department of Chemistry
University of Science & Technology of China
Hefei, Anhui, China, 230026

REVIEWERS' COMMENTS:

Reviewer #1 (Remarks to the Author):

The revised manuscript has been significantly improved, and all the points raised by the reviewers have been addressed properly. I think this research work would be suitable for Nature Communication after the following mistakes were corrected.

- 1) Line 19, "unpresendent" should be "unprecedented".
 - 2) Line 23, "singe" should be "single".
 - 3) Line 29, "ortho-substituents" should be "ortho-substituents".
 - 4) Line 107, "91% ee" was not consistent with entry 5 (92% ee) in Table 2.
 - 5) Line 145, "reactedwith" should be "reacted with".
 - 6) Line 153, "88% ee" was not consistent with that in Fig. 4. (87% ee).
 - 7) Line 166, "a 1:1.5 ratio mixture of 6a and 6c" was not consistent with that in Fig. 5b. (6a : 6c = 1.5:1).
 - 8) Line 210, "ccomplex" should be "complex".
 - 9) I am not able to download the supporting information, and I am not sure the ratio of mixtures in Fig 5 a and Fig 5b was determined by ¹HNMR of crude products or isolated yields. Hence, the ¹HNMR of the crude mixtures in Fig 5 a and Fig 5b should be provided.
- The authors should check the whole manuscript for other similar mistakes.

Reviewer #2 (Remarks to the Author):

This is a revised manuscript on a rhodium-catalyzed coupling reaction of silafluorenes with acid anhydrides.

The authors diligently addressed the reviewers' concerns, and the revisions they made are appropriate. I consider the paper is ready for publication without further change.

We would like to thank all the reviewers for your time and efforts in evaluating our manuscript (Manuscript ID: NCOMMS-20-12289). We have made all of the requested changes, which are listed point-by-point in this letter.

Reviewer 1:

1. **Comment:** The revised manuscript has been significantly improved, and all the points raised by the reviewers have been addressed properly. I think this research work would be suitable for Nature Communication after the following mistakes were corrected.

Answer: We thank the reviewer's comments.

2. **Comment:** Line 19, "unpresendent" should be "unprecedented".

Line 23, "singe" should be "single".

Line 29, "ortho-substituents" should be "ortho-substituents".

Line 107, "91% ee" was not consistent with entry 5 (92% ee) in Table 2.

Line 145, "reactedwith" should be "reacted with".

Line 153, "88% ee" was not consistent with that in Fig. 4. (87% ee).

Line 166, "a 1:1.5 ratio mixture of 6a and 6c" was not consistent with that in Fig. 5b. (6a : 6c = 1.5:1).

Line 210, "ccomplex" should be "complex".

Answer: (a) The typos have been corrected.

(b) "91% ee" has been changed to "92% ee"; "88% ee" has been changed to "87% ee"; "a 1:1.5 ratio mixture of 6a and 6c" has been changed to "a 1.5:1 ratio mixture of 6a and 6c";

3. **Comment:** I am not able to download the supporting information, and I am not sure the ratio of mixtures in Fig 5a and Fig 5b was determined by ¹HNMR of crude products or isolated yields. Hence, the ¹HNMR of the crude mixtures in Fig 5a and Fig 5b should be provided. The authors should check the whole manuscript for other similar mistakes.

Answer: The ratio of the mixture in Fig 5a and Fig 5b was determined by the ¹H NMR

of crude products. The ^1H NMR of crude mixtures of **6gg** and **6gg'** in Fig 5a; **6m** and **6b** in Fig 5a; **6b** and **6c** in Fig 5a; **6a** and **6c** in Fig 5b have been added.

Reviewer 2:

1. Comment: This is a revised manuscript on a rhodium-catalyzed coupling reaction of silafluorenes with acid anhydrides. The authors diligently addressed the reviewers' concerns, and the revisions they made are appropriate. I consider the paper is ready for publication without further change

Answer: We thank this reviewer's comments.

We thank all reviewers for their valuable comments that we believe substantially improved the quality of our manuscript.

With best regards,

Zhenhua Gu, Ph.D.
Professor of Chemistry
Department of Chemistry
University of Science & Technology of China
Hefei, Anhui, China, 230026